# Existing and Emerging Metabolomic Tools for ALS Research

**DOI:** 10.3390/genes10121011

**Published:** 2019-12-05

**Authors:** Christine Germeys, Tijs Vandoorne, Valérie Bercier, Ludo Van Den Bosch

**Affiliations:** 1Department of Neurosciences, Experimental Neurology, and Leuven Brain Institute (LBI), KU Leuven—University of Leuven, 3000 Leuven, Belgium; christine.germeys@kuleuven.vib.be (C.G.); tijs.vandoorne.work@gmail.com (T.V.); valerie.bercier@kuleuven.vib.be (V.B.); 2VIB, Center for Brain & Disease Research, Laboratory of Neurobiology, 3000 Leuven, Belgium

**Keywords:** amyotrophic lateral sclerosis, motor neuron, metabolomics, mass spectrometry, energy metabolism, metabolic dysfunction

## Abstract

Growing evidence suggests that aberrant energy metabolism could play an important role in the pathogenesis of amyotrophic lateral sclerosis (ALS). Despite this, studies applying advanced technologies to investigate energy metabolism in ALS remain scarce. The rapidly growing field of metabolomics offers exciting new possibilities for ALS research. Here, we review existing and emerging metabolomic tools that could be used to further investigate the role of metabolism in ALS. A better understanding of the metabolic state of motor neurons and their surrounding cells could hopefully result in novel therapeutic strategies.

## 1. Introduction

Amyotrophic lateral sclerosis (ALS) is a fatal neurodegenerative disorder characterized by the selective loss of both upper and lower motor neurons in the motor cortex, brainstem and spinal cord. The death of motor neurons leads to the denervation of neuromuscular junctions and results in progressive muscle atrophy. In typical cases, the death of the patient is due to respiratory failure, and occurs within three to five years of diagnosis [1]. In 10% of ALS cases, the disease is caused by a genetic defect, which is mainly inherited via an autosomal dominant Mendelian pattern. A genetic analysis of these familial ALS (fALS) cases showed that mutations in superoxide dismutase 1 (*SOD1*), in TAR DNA binding protein (*TARDBP*) encoding TDP-43, and in fused in sarcoma (*FUS*), as well as hexanucletoide repeat expansions in chromosome 9 open reading frame *72* (*C9ORF72*) are the most common genetic causes of ALS [2]. While more than half of fALS patients carry mutations in these genes, mutations in more than 25 genes have been linked to ALS [3]. Despite the discovery of a large set of ALS-associated genes, these findings have not resulted in effective therapeutic strategies to this day. Furthermore, the pathological mechanisms involved in ALS are still incompletely understood.

Growing evidence suggests the involvement of aberrant energy metabolism in ALS etiology (as we recently reviewed [4]). Motor neurons are known to be particularly vulnerable to energetic stress. Their vulnerability has been mainly attributed to their long neurites, which require a lot of energy to insure action potential propagation to their synaptic targets [5]. In addition, axonal transport is an energy-demanding process that is essential to provide synapses with the necessary proteins and sufficient energy, as well as to mediate signaling back to the cell body and to ensure clearance of detritus [6]. The metabolic vulnerability of motor neurons in ALS is further underlined by the initial degeneration of fast-fatigable motor neurons, which require higher peak needs of ATP compared to the slow motor neurons, less prone to neurodegeneration in this context [7]. While glucose is the main energy substrate of the central nervous system [8], it is still unclear to which extent glucose is processed before it is used by neurons to meet their energy demand [9]. In vitro and ex vivo studies have shown that neurons oxidize other non-glucose substrates such as glutamate, a process which has been suggested to protect neurons from excitotoxicity [9,10,11]. Additionally, neurons are also able to process ketone bodies to maintain energetic homeostasis [12,13,14]. As opposed to glia, neurons are suggested to be primarily oxidative, meaning that glycolysis is followed by the full oxidation of glucose or of its metabolites by the tricarboxylic acid (TCA) cycle and the electron transport chain taking place in the mitochondria [15,16]. Due to their metabolic vulnerability, we believe that the selective loss of motor neurons in ALS patients could be the result of a dysregulated energy metabolism in these cells.

This hypothesis is supported by clinical and preclinical evidence. Indeed, both fALS and sporadic ALS (sALS) patients exhibit decreased weight and fat depots, which could be the result of a combined decrease in energy uptake, for example, due to a loss in appetite, and an increase in energy expenditure [17,18,19,20]. While dysbalanced energy homeostasis is an early event in ALS rodent models [21,22,23,24], it remains to be determined whether this is a primary or secondary event in ALS patients [4].

In addition to the systemic metabolic changes, changes have been observed at different levels of cellular energy metabolism. First, cellular energy homeostasis appears to be impaired in ALS. Indeed, the activity of AMP-activated protein kinase (AMPK) was shown to be elevated in motor neurons of ALS patients [25]. This kinase is an important player in cellular energy homeostasis as it acts as a cellular energy sensor. Active AMPK results in an increase in catabolic and a decrease in anabolic pathways [26]. Moreover, 5-aminoimidazole-4-carboxamide-1-β-d-ribofuranoside (AICAR)-mediated activation of AMPK in NSC34 motor neuron-like cells resulted in cytoplasmic TDP-43 mislocalization, linking energy depletion to TDP-43 pathology [25]. Secondly, mitochondrial dysfunction is known to be an ALS hallmark. These organelles play a crucial role in cellular homeostasis due to their role in energy supply, as they are the site where oxidative phosphorylation takes place, and due to their involvement in calcium homeostasis and apoptosis [27]. Mitochondrial dysfunction has been studied extensively in ALS because *SOD1*, encoding superoxide dismutase 1 which is also localized in mitochondria, was the first gene in which ALS-causing mutations were discovered [28]. Subsequently, mitochondrial changes have been reported in ALS patients and in murine, cell line, and human induced pluripotent stem cell (iPSC)-derived SOD1-ALS models but also in TDP-43 and FUS models [24,25,26,29,30,31,32,33,34,35,36,37]. In our opinion, the latter indicates that mitochondrial dysfunction could represent a general hallmark of ALS rather than being attributed to mutations in *SOD1*. Yet, we recently showed that mitochondrial dysfunction is absent in FUS-ALS human motor neurons [9]. Thirdly, evidence points to a defective carbohydrate metabolism in ALS. Studies using [^18^F]-2-fluoro-2-deoxy-D-glucose positron emission tomography (PET) reported that cortical glucose uptake is reduced in ALS patients while local glucose uptake seems to be increased in both the cervical and dorsal spinal cord [38,39,40]. Furthermore, cortical hypometabolism is suggested to be an early diagnostic event that can be linked to the severity of the disease [38,41]. Similarly, in vivo capillary imaging in mutant SOD1^G93A^ mice indicates that glucose uptake in the spinal cord is increased presymptomatically but declines as the disease progresses [42]. Microarray analysis and proteomics studies also provided evidence for a disturbed carbohydrate metabolism in ALS. Indeed, several studies explored the downregulation of glycolysis components in ALS patients, including the key enzymes phosphoglucomutase 2 like 1 and phosphoglycerate kinase [23,43]. Consistently, 6-phosphofructo-2-kinase/fructose-2,6-biphosphatase 3 (PFKFB3) mRNA was found to be reduced in the post-mortem cortex of ALS patients [44]. Finally, in addition to the carbohydrate metabolism, also lipid metabolism is suggested to be affected in ALS. Dysregulated lipid metabolism has mainly been attributed to skeletal muscle tissues which upregulate their lipid-based energy metabolism early in the course of the disease [45]. Nevertheless, research focusing on the role of lipid metabolism in motor neuron toxicity has been instigated by the observation that increased levels of lipid-derived oxidative pathway intermediates could contribute to neuronal oxidative stress [46,47,48,49]. As glucose metabolism seems to be downregulated, it is hypothesized that other energy substrates are used to meet the energy demand in motor neurons. In support of this, lipid catabolism is found to be increased in ALS mice and ketone bodies are elevated in the cerebrospinal fluid of ALS patients [50,51,52,53]. Moreover, proliferator-activated receptor gamma coactivator 1 alpha (PGC1α) and stearoyl-CoA desaturase 1 (SCD-1), both involved in fatty acid metabolism, are reported to be downregulated in ALS mouse models and ALS patients [54,55,56,57,58].

Interestingly, the lipid metabolism also provides a close link between energy and epigenetic regulation, another process which is increasingly acknowledged to be implicated in ALS pathophysiology (for a review, see [59]). For instance, histone acetylation, a regulator of chromatin folding and hence transcription factor accessibility, is highly dependent on the levels of acetyl-CoA, a product of the TCA cycle and fatty acid oxidation and the substrate for de novo fatty acid synthesis [60,61]. Moreover, histone acetyl transferases (HATs) and histone deacetylases (HDACs), the main regulators of histone acetylation, have been shown to be involved in the regulation of energy metabolism [62,63,64,65,66]. Therefore, changes in metabolic homeostasis could be closely related to changes at the epigenetic level. Furthermore, metabolic changes have recently been suggested to result from changes in the microbiome. The gut-microbiome is a major source of low-molecular-mass metabolites that are able to modulate metabolic, transcriptional and epigenetic processes upon crossing the blood brain barrier [67]. Evidence for the involvement of a disturbed microbiome in ALS is provided by several studies that reported a dysbiotic microbiome and an increased gut permeability in ALS mouse models and patients [68,69,70,71].

Although published evidence suggests a connection between a disturbed energy metabolism and ALS etiology, the use of advanced technology to investigate energy metabolism in ALS is necessary to discriminate causation from correlation. Indeed, it remains unclear whether the observed metabolic defects discussed above are a primary event or secondary to other defects such as aberrant epigenetic regulation or a disturbed microbiome. We recently applied state-of-the-art metabolic tools in ALS to investigate the effect of ALS-causing mutations in *FUS* on the metabolic state of human motor neurons [9]. With the advent of several new technologies and models available to unravel the metabolic differences between healthy and diseased motor neurons and their microenvironment, we here provide an overview of existing and emerging metabolic technologies to guide further investigation on the exact role of metabolism in ALS.

## 2. Existing Tools to Measure Energy Metabolism

A number of metabolic tools are already available and can be applied to measure the metabolic state in ALS models (Table 1). Published use of these tools resulted in an increased, but still limited, knowledge on the metabolic state in ALS. Here, we discuss the available metabolic toolkit and its application in ALS research.

### 2.1. RNA Sequencing of Flux Generating Enzymes

One way to indirectly investigate metabolism is by measuring changes in the abundance of flux generating enzymes. This can be achieved by using RNA sequencing to gather information about metabolic pathways by detecting changes in the RNA expression levels of enzymes and cofactors [72]. As an example, it was shown that PFKFB3 mRNA levels are reduced in the post-mortem cortex of ALS patients [44]. Because PFKFB3 is known to be a positive regulator of glycolysis, this finding supports a downregulation of glucose metabolism in ALS and is consistent with the results obtained in PET and proteomic studies [23,38,39]. Moreover, we recently showed that HDAC inhibition led to the amelioration of the motor phenotype in a FUS-ALS mouse model [66]. This effect was linked to an increased expression of genes associated with glycolysis, the pentose phosphate pathway, and lipid transport, and to a decreased expression of genes related to fatty acid metabolism, partly restoring mRNA levels [66]. However, since changes at the transcriptome level do not necessarily translate into changes in the proteome or into functional changes, a fortiori not in energy metabolism, expression studies should be confirmed with functional metabolic studies [73].

### 2.2. Extracellular Metabolic Flux Analysis

Another indirect way to measure cellular metabolism, is the analysis of cellular respiration. This can be achieved by measuring the oxygen consumption rate and extracellular acidification rate as a readout for mitochondrial respiration and for lactate production in vitro, via the use of tools such as the Seahorse XF bioanalyzer and the Oxygraph-2k [74,75]. In both cases, oxygen and protons can be measured in an enclosed chamber, where the Seahorse system is advantageous in that the microchamber is incorporated in the cell culture plate while the Oxygraph assay is performed on the cell suspension. Therefore, the Seahorse system allows easy respiratory measurement of cultured cells such as motor neurons or fibroblasts and can even be applied to muscle fibers [23,76,77]. Both allow the measurement of basal oxygen consumption but also other aspects of mitochondrial respiration by using different inhibitors. For instance, by adding ATP synthase inhibitor oligomycin to the culture medium, the oxygen consumption rate will drop, reflecting the oxygen consumption related to ATP generation. Carbonilcyanide p-triflouromethoxyphenylhydrazone (FCCP), a mitochondrial uncoupler, increases the oxygen consumption to the maximal respiratory capacity, indicative for the reserve electron transporting capacity of mitochondria. Finally, complex III inhibitor antimycin-A can be used to shut down the electron transport chain, allowing the measurement of proton leakage as the difference in oxygen consumption rate after oligomycin and antimycin-A injection. Several studies have used this approach to investigate changes in mitochondrial respiration related to different ALS-associated genes. For example, a decreased oxygen consumption rate and coupling efficiency was reported, together with upregulation of most energy-transducing enzymes, including components of the fatty acid oxidation, of the TCA cycle, of glycolysis and of oxidative phosphorylation, in motor neurons of *SOD1^G93A^* mice [23]. In line with this, motor neuron-like cells carrying a *SOD1^G93A^* mutation showed the most prominent decrease in mitochondrial respiration and coupling efficiency compared to other *SOD1* mutations [78]. Furthermore, a decreased spare respiratory rate, increased extracellular acidification rate, proton leak and decreased ATP levels together with higher mitochondrial enzyme complexes II+, III and complex IV activities, suggested that mutations in *VCP* affected the mitochondria’s ability to produce ATP in patient and mouse fibroblasts [76]. *C9ORF72* hexanucleotide repeat expansions in fibroblasts were also linked to mitochondrial dysfunction, showing increased oxygen consumption and mitochondrial hyperpolarization associated with increased reactive oxygen species (ROS) and ATP content compared to controls when inducing oxidative metabolism [79]. In contrast, fibroblasts expressing mutant *TARDBP* did not show any changes in the oxygen consumption rate in this condition [79]. In addition, fibroblasts from sALS patients showed a different response to ageing compared to controls as an increase in uncoupled mitochondrial respiration, but no decrease in glycolysis, no increase in the oxygen consumption rate or in extracellular acidification rate were observed [80]. Finally, we recently showed that ALS-causing mutations in FUS do not affect cellular respiration in human motor neurons [9]. Altogether, these studies illustrate the value of the Seahorse XF analyzer and its ability to obtain reliable information about mitochondrial respiration and extracellular acidification.

### 2.3. Intracellular Metabolic Flux Analysis

Intracellular flux analysis is a valuable tool to directly quantify cellular metabolism. A longstanding and robust approach to measure intracellular metabolic fluxes is achieved by performing radioactive tracing, as measuring radioactivity gives direct information about the metabolic flux through the pathway of interest. In this assay, metabolic energy substrates are labeled using radioactive isotopes. Common radioactive tracers in the assessment of metabolism are ^3^H or ^14^C, as their use results in the generation of radiolabeled H_2_O and CO_2_, respectively, which allows easy detection (for a review, see [81]). For example, glycolysis can be measured using D-[5-^3^H(N)]-glucose as tracer. When glycolysis is active, ^3^H_2_O will be produced when 2-phosphoglycerate is converted to phosphoenolpyruvate by the enolase enzyme. As a result, the flux rate of glycolysis can be assessed by measuring the produced ^3^H_2_O over time using a scintillation counter [81].

In the context of ALS, this method could be used to compare healthy with diseased motor neurons in culture. We recently used radioactive flux tracing to show energy substrate-specific metabolic rewiring during motor neuron differentiation [9]. Moreover, we investigated potential metabolic defects caused by ALS-related mutations in *FUS* and showed that various mutations did not affect the energy metabolism of patient-derived motor neurons [9]. However, our results do not rule out that motor neuron-extrinsic metabolic alterations could contribute to or could cause motor neuron degeneration [82,83,84]. While we showed that potential metabolic defects resulting from FUS-ALS are not motor neuron-intrinsic, it is possible that metabolic dysfunction in neighboring cells contributes to motor neuron degeneration in ALS patients or models.

### 2.4. Metabolomics

The measurement of abundance for various metabolites is an unbiased way to explore the metabolic condition linked to health or disease. Metabolites are more challenging to measure when compared to peptides, as they are smaller (0.5–2 kDa) and often lack common building blocks [85]. Metabolomics tries to overcome these challenges by providing a tool for high-throughput analysis of these biomolecules. The most commonly used analytical techniques are mass spectrometry (MS) combined with liquid or gas chromatography (LC-MS and GC-MS) and nuclear magnetic resonance (NMR), which are both supplementary and complementary to one another. Other available analytical tools are capillary electrophoresis-mass spectrometry (CE-MS) and ion mobility-mass spectrometry (I-MS) [86,87].

Separation of polar/non-polar metabolites and charged metabolites by, respectively, LC- and GC-MS is followed by ionization [88,89], typically by electrospray ionization (ESI) and electron impact (EI) ionization [90]. Detection of metabolites by the numerous available MS techniques is based on separation according to specific mass-to-charge ratio (*m*/*z*). Mass analyzers resolve the created ions either in a time-of-flight tube or in an electromagnetic field and can be used in a single (MS) or tandem (MS/MS) configuration. For targeted metabolomics, the MS methods of choice are single quadrupole, quadrupole ion trap, triple quadrupole and orbitrap [91]. Single quadrupole is the simplest MS method and is mainly used in the single-ion monitoring mode, where the parameters can be adjusted to filter and select for one specific *m*/*z* and allow detection of a specific biomolecule [92]. The quadrupole ion trap works with a similar principle as the single quadrupole, using an electric field to select ions with a specific *m*/*z* [93]. The triple quadruple is able to achieve higher sensitivity by performing selected reaction monitoring experiments [94]. Here, selection of specific precursor ions is performed in the first MS, similarly to single quadrupole. Subsequently, fragmentation by collisions with an inert gas occurs in the collision cell and is followed by detection of specific fragment ions in the third quadrupole. Finally, the orbitrap is the most recent MS-based technique, in which ion oscillations in an electrostatic field are used to measure *m*/*z* providing high acquisition speed together with high mass resolution and accuracy [95].

In addition to MS, NMR analyses have also been used to perform metabolite analysis [96]. Using this method, samples can be analyzed directly with minimal manipulation while measuring a wide range of small molecules simultaneously. NMR does not require a complex sample preparation, shows high levels of experimental reproducibility and is non-destructive. However, the major drawbacks of using NMR to quantify energy metabolism are the poor sensitivity and the spectral complexity [97]. Because of this, MS remains the method of choice for metabolomics.

Although promising, few studies have used metabolomics to explore energy metabolism in the context of ALS. Nevertheless, metabolomics showed that introducing mutant *SOD1* in NSC34 motor neuron-like cells resulted in increased glycolysis [98]. Metabolomic analysis in NSC34 cells and C8-D1A astrocyte-like cell co-cultures also showed potential disruption of the TCA cycle and glutamate metabolism under oxidative stress conditions when overexpressing wild-type or mutant *SOD1* [99]. Furthermore, metabolomics could play a crucial role in the search for biomarkers in ALS. Indeed, metabolomics was used to determine the presence of possible biomarkers in cerebrospinal fluid or plasma from ALS patients [52,53,100,101,102,103,104]. These studies did not identify a single outstanding biomarker but were able to find distinctive metabolic profiles in ALS patients.

### 2.5. Stable Isotope Tracing

Conventional metabolomic techniques, as described above, measure the abundances of metabolites in a given sample but can also be combined with stable isotope tracing to map the detected biomolecules to metabolic networks and pathways. Unlike radioactive isotopes, stable isotopes do not emit energy in the form of ionizing radiation and are detected based on their mass-difference. The most commonly used isotopes in metabolomics are ^2^H, ^13^C and ^15^N [105,106,107]. When the stable isotope enrichment is stable over time, isotopic steady-state is reached and analysis of the labeling patterns can give information about relative pathway activities, about changes in the contribution of labeled nutrients to pathways and about the production of different metabolites [106]. Analysis of non-steady state data is also possible, although this requires a different approach [108]. Furthermore, isotopic labelling also offers a way to indirectly measure metabolic flux by analyzing the metabolite levels before metabolic steady-state is reached [109].

For example, we used ^13^C_6_-glucose and ^13^C_3_-lactate fate mapping to show that during motor neuron differentiation, iPSCs undergo a prooxidative metabolic switch fueled by lactate oxidation and rewire metabolic routes to import pyruvate into the TCA cycle in an energy substrate-specific way. In addition, we showed that major catabolic pathways were not affected in FUS-ALS patient derived motor neurons [9]. Also in the context of proteomics, isotopic labelling of amino acids, referred to as labeling by amino acids in cell culture (SILAC), has been used to study protein synthesis and protein interactions [110]. Application of SILAC on cellular models of ALS showed a disturbed interaction of mutant FUS with survival motor neuron protein (SMN) and U1-snRNP, and the interaction of TDP-43 with numerous proteins such as ras GTPase-activating protein (G3BP), polyadenylate-binding protein 1 (PABPC1) and eukaryotic translation initiation factor 4A1 (eIF4A1) and hnRNP [111,112,113].

Several studies suggested that energy metabolism is disturbed in ALS but to date, the contribution of changes in energy metabolism to ALS pathology remains unclear. Metabolomics thus offers a way to analyze the metabolome and represents a promising tool for ALS research as it could lead to a better understanding of the suggested failure in energy metabolism. However, the conventional metabolomic tools described above have limitations to consider. For example, MS-based metabolomics, as discussed above, requires a significant amount of starting material and fails to discriminate potential differences between individual cells or cellular compartments.

## 3. Emerging Metabolomic Tools for ALS Research

In this section, we present a snapshot of emerging tools in this rapidly progressing omics field that could be of particular interest to ALS research (Table 1). To our knowledge, these techniques have not, or only very scarcely, been applied in ALS. We believe that applying these tools could lead to a better understanding of the metabolic defects in ALS.

### 3.1. Untargeted MS-Based Metabolomics

The rapid growth of the metabolomics field relies on more advanced, high-throughput analytical platforms together with new tools for data processing [114]. Recently, technical progress led to the development of untargeted metabolomics, which does not focus on a predefined set of metabolites but rather analyzes all known and unknown metabolites present in the sample. The strength of untargeted metabolomics is its unbiased approach to the metabolic state. Both LC-MS and GC-MS techniques are amenable to untargeted metabolomics [89,115], although LC-MS is often preferred over GC-MS as GC-MS has lower coverage due to the fact that major classes of metabolites cannot be made sufficiently volatile [116]. Following sample separation and ionization as discussed above, data acquisition can be obtained by either data-dependent acquisition (DDA) or data-independent acquisition (DIA), which both use tandem mass spectrometry (MS/MS) to acquire data [117,118]. In the first common step, ions are separated by performing a full MS1. Next, in DDA, the precursor ions are ranked based on their intensity where the most intense ions are then isolated and fragmented to acquire their MS2 spectrum. Limitations of this data acquisition technique are its selection bias towards more abundant precursor ions and the undefined MS2 spectral quality. These limitations make DDA less suitable for untargeted metabolomics. Using DIA, all precursor ions within a selected window are being isolated to acquire MS2 spectra. Therefore, DIA is able to overcome the selection bias present in the DDA approach. However, with DIA the link between MS1 and MS2 spectra is missing, which complicates the analysis, but MS1 and MS2 spectra can be linked again by matching their retention time, mass, and drift time [119]. After data acquisition by either DIA or DDA, precursor and corresponding fragment ions can be identified by comparing their spectra and retention time with databases for metabolite assignments [120].

Some issues arise using untargeted metabolomics regarding the compound identification and the contextualization of metabolites into biological pathways [121]. An analytical tool that has been proposed to improve metabolite identification is I-MS, a gas-phase based method in which size-to-charge separation is based on cross-section charge state of the metabolite and its interactions with the buffer gas [87]. With this analytical method, chemical modifiers can be added to facilitate the identification of position isomers [122,123]. Despite the proposed solutions, the difficulty of metabolite identification remains a limitation of untargeted metabolomics. Nevertheless, the metabolome coverage of untargeted metabolomics is significantly higher compared to targeted metabolomics. Untargeted metabolomics is able to produce a relative quantification, which would be useful when comparing healthy and ALS samples. The detected changes could subsequently be confirmed by absolute quantification using targeted metabolomics. Indeed, untargeted metabolomics should not replace targeted metabolomics but these techniques can complement each other and would ideally be performed in parallel. Blasco and colleagues were the first to apply untargeted metabolomics on cerebrospinal fluid of ALS patients which, resulted in the generation of a specific metabolic profile for ALS [124]. This study highlights the opportunity to apply untargeted metabolomics in the search for ALS biomarkers but also in the search for the role of metabolism in ALS etiology.

### 3.2. In Vivo Metabolic Tracing

The majority of metabolomic tracing studies are performed on in vitro models, in which one can easily apply stable isotopes and efficiently extract metabolites. ALS can be modeled by primary cultures of rat or mouse motor neurons, but also by differentiated cultures obtained from patient-derived iPSCs [125,126,127]. ALS studies have shown that microglia, oligodendrocytes and astrocytes contribute to motor neuron degeneration, suggesting that the field would benefit from in vivo metabolomic analyses [128,129,130,131]. Indeed, the metabolism of motor neurons is believed to be affected by their microenvironment. An example of this is the astrocyte-neuronal lactate shuttle hypothesis, which postulates that lactate could be provided to neurons by astrocytes [82,83]. Oligodendrocytes are also known to support the metabolic state of motor neurons by providing lactate [132]. Moreover, it was shown previously that mitochondrial defects in muscles resulted in degeneration of neuromuscular junctions [133]. Furthermore, recent discoveries showed that defects in adenosine, fructose and glycogen metabolism in induced astrocytes from C9ORF72-ALS and sALS patients were linked to a higher susceptibility to adenosine-induced toxicity and defective metabolic flexibility in motor neurons and astrocytes respectively [134,135].

A suitable model for in vivo metabolomics in ALS is the rodent model, given the valuable information gained from the use of previously established mouse models. While the mutant SOD1^G93A^ mouse model is best characterized, other SOD1, as well as TDP-43, FUS, UBQLN2, VAPB and VCP mouse models have been generated (for a review, see [136]). Stable isotope tracers could, for example, be applied to these mice via a cannula which is surgically applied to the right jugular vein [137]. After incubation, the metabolites could be extracted from the blood or from any tissue of interest to perform metabolomics analysis. Another suitable option for in vivo metabolomics is the zebrafish model (for a review, see [138]). Indeed, this vertebrate model is well established and has been used extensively in the past decade to study ALS [129,139,140,141,142,143,144,145], as it has an added value because of the ease of genetic manipulation and the translucency of the embryonic and larval stages, allowing access for probing molecular mechanisms in vivo [146,147,148,149]. Studies describing the use of methods to investigate metabolic processes in this model organism have been published in recent years, showing that techniques such as GC-MS and LC-MS, ^1^H NMR-based metabolomics, high-resolution magic-angle spinning nuclear magnetic resonance NMR spectroscopy (HRMAS-NMR), shotgun MS and DIA-based Sequential Window Acquisition of All Theoretical Mass Spectra (SWATH-MS) protein profiling assay, can be successfully used [150,151,152,153,154,155]. Moreover, systemic administration of isotopic tracers could be achieved via injection [156]. Aside from animal models, an advantage of stable isotope tracing is that it is also suitable for use in humans. Unlike radioactive isotopes, stable isotopes do not emit ionizing radiation and can be safely used in human participants. For example, deuterium tracing provides an accessible method for in vivo tracing of lipid metabolism as it can be applied systemically in the form of heavy water (^2^H_2_O) [105]. Moreover, other stable isotope tracers can be introduced in the systemic circulation via intravenous application [157]. Measurement of biomolecules can be performed on blood or tissue samples and the analysis can be done via LC-MS, commonly used for blood, urine, or tissue extracts [158]. This detection method is suited for biomolecules with a wide range of polarity and it does not require derivatization which speeds up the pretreatment process. In addition, GC-MS and NMR-MS have also been used for serum-based analysis [159,160]. MS allows for the relative abundances of metabolites to be measured and can be used to calculate the tracer (labeled metabolite) to tracee (unlabeled metabolite) ratio (TTR), providing information about the systemic metabolite (tracee) kinetics.

The ability to perform stable isotope tracing in vivo offers great opportunities for ALS research, as it is a complex disease in which motor neurons are known to be affected by surrounding cells [128,130,161]. Given that astrocytes and oligodendrocytes provide lactate to motor neurons, which is assumed to contribute to neuronal survival, a decrease in glycolysis in these glial cells could result in a decrease in the oxidative phosphorylation in motor neurons due to a decrease in lactate as energy substrate [82,83,132,162]. Moreover, the findings linking astrocyte and motor neuron toxicity in C9ORF72-ALS and sALS to the metabolic dysregulation of astrocytes together with the detrimental effects of mitochondrial defects in muscles on neuromuscular junctions underline the possibility of motor neuron extrinsic metabolic defects in ALS [133,134,135]. Indeed, in vivo metabolomics potentially capture the motor neuron microenvironment in a better way compared to in vitro metabolomics and could therefore lead to new insights into the metabolic defect in ALS.

### 3.3. Single Cell Metabolomics

Another tool that can incorporate the microenvironment of motor neurons in the analysis of the metabolism is co-culture, either in 2D or 3D. In these conditions, multicell or tissue-scale metabolomics could produce misleading data about the conditions of a cell since it yields averaged data. However single cell metabolomics could enable us to look at the metabolic state of the different cell types and subtypes separately. Moreover, as it has been suggested that motor neurons could suffer from non-cell autonomous toxicity, single cell metabolomics could provide a way to investigate the metabolic state of all cells present in the brain and spinal cord in parallel, instead of focusing on the motor neurons.

One of the most critical steps in single cell metabolomics is the cell sampling. The methods used for traditional metabolomics are not applicable to single cell applications because a higher amount of input material is required and single cell methods are often too time consuming, affecting the metabolic state of the cells. Two solutions have been proposed to deal with this issue. The first one is the use of nanodevices to separate the cells. An example of a nanodevice is the micropipettes, which can be coupled to automated systems targeting different cell types based on visual cues. After the selection, the target cell can be picked and manually guided by a robotic system [163]. Nevertheless, micropipettes are rarely used because this technique highly deforms cells and does not allow for simultaneous profiling of a large number of cells, a requirement for overcoming intrinsic noise and to ensure data quality [164]. Secondly, microfluidic arrays represent another tool for single cell isolation that is suitable for metabolomics. In that case, cells are isolated by physically confining them within microfluidic structures, where the most extensively used ones are nano-liter wells, droplets and valve-controlled channels [165,166,167]. For example, a commonly used microfluidic array system uses a branched array that consists of polydimethylsiloxane traps, allowing the isolation of one or two cells in a gentle manner [168]. Once isolated, metabolites can be extracted and the metabolome can be measured, although some challenges related to the single cell measurement could arise. Indeed, because of the small sample size, it becomes difficult to obtain a good resolution and a sufficient sensitivity, both needed to detect the small concentrations. Nevertheless, mass spectrometry is well suited for single cell analysis due to its relatively high resolution and sensitivity [169]. Several analytical methods for single-cell metabolomics have been suggested. A widely used method reported to be suitable for single-cell metabolomics is matrix-assisted laser desorption ionization (MALDI) [170], in which the samples are applied to a plate where they solidify and form crystals. The plate can be loaded into a mass spectrometer where the sample is evaporated by a UV beam in a vacuum space, which results in the ionization of the biomolecules in the sample. After ionization, MS results in a spectrum for each metabolite present. MALDI is able to achieve sufficiently high sensitivity for the application in single cell metabolomics but still encounters some limitations. The vacuum condition used for the ionization of the samples does not recapitulate the natural environment of the cell. Furthermore, the matrix used with MALDI results in molecular signals in the low molecular ranges (<0.5 kDa), where most small biomolecules are also found. Live single-cell mass spectrometry (LSC-MS) is able to overcome these limitations because this method allows the cells to remain in their natural environment or in a preferable medium until seconds before the MS analysis [171]. A single cell, or single cell content, is captured by a metal-coated nanospray tip attached to a micromanipulator. After adding a standard ionization solvent, the content can be brought into a nano-ESI on a mass spectrometer and is then followed by measurement of the molecular content. This method is able to detect 100 to 1000 molecular peaks within minutes. However, LSC-MS is low-throughput and results in the presence of intrinsic noise because the simultaneous analysis of several single cells using this approach is not possible. Another matrix-free alternative is the widely used method of Secondary Ionization Mass Spectrometry (SIMS) [172]. SIMS uses a focused beam of primary ions to bombard the sample surface, creating secondary ions from the analyte, which can be followed by metabolite mass analysis using time-of-flight [173]. SIMS is especially suitable for lipidomic analysis and has previously been combined with MALDI to obtain information about small metabolites, lipids and peptides from the same cell. SIMS offer a higher spatial specificity than MALDI but at the expense of molecular fragmentation, resulting in lower chemical specificity [174]. Finally, laser ablation electrospray ionization (LAESI) offers a matrix-independent approach for in situ metabolite analysis of tissue and single cells, particularly interesting for its application on cultured brain and spinal cord tissue [175]. LAESI uses a mid-infrared laser to ablate the sample in situ, generating gas phase particles which are subsequently ionized by an ESI source for MS analysis. LAESI is a relatively new methods but has the potential to contribute to single-cell metabolomics as the technology improves.

Most of these single cell studies use shotgun-like approaches, which means that MS is not preceded by a separation step. This approach makes it more difficult to distinguish isomeric metabolites. In addition, random and systematic errors could arise due to matrix effects like ion suppression or enhancement due to the simultaneous ionization of multiple analytes. Single cell metabolomics can nevertheless lead to new information about the metabolic conditions of motor neurons and surrounding cell, potentially lost in the bulk measurement of traditional metabolomics.

### 3.4. Metabolic Compartmentalization

The abovementioned techniques measure the average metabolic activity of the cell. However, recent knowledge has been gained about subcellular compartments suggesting each could harbor specific metabolic activities [176]. Spatial metabolic compartmentalization is believed to have evolved in order to overcome challenges such as the low efficiency or specificity of some metabolic enzymes, or the cellular toxicity exerted by several intermediate metabolites [177]. Compartmentalization of metabolic pathways can be obtained by the co-localization of metabolic enzymes through the interactions between enzymes, between enzymes and scaffold proteins, but also through encapsulation of the enzymatic pathways in subcellular compartments [178,179].

In the context of energy metabolism, one of the most obvious subcellular compartments are mitochondria. Mitochondria harbor both the TCA cycle and fatty acid oxidation, and both their morphology and function are disturbed in ALS patients and models [24,29,32,33,34,35,36,37]. Despite the interest in mitochondrial metabolism, few studies focused on the selective measurement of the mitochondrial metabolome in the context of ALS [180]. While separating the mitochondrial from the cytosolic metabolome remains challenging, methods have been generated to selectively study the mitochondrial metabolome. The discussed methods are based on the isolation of mitochondria followed by stable isotope tracing. To start, mitochondria can be isolated from tissue or cultured cells by differential centrifugation [181,182]. The quality of the mitochondrial suspension is subsequently evaluated based on the respiratory control ratios, which are calculated by dividing the rate of oxygen consumption in the presence of ADP by the oxygen consumption in the presence of oligomycin, both measured by a Clark-typed electrode [182]. After quality control, the mitochondrial suspension is incubated with stable isotope tracers, followed by sample preparation and MS analysis. The preferred analytical tool is GC-MS because the sensitivity and coverage of this method were optimized for TCA intermediates [183]. This approach is amenable to the study of the mitochondrial metabolome but still represents a time-consuming method and a lot of material is necessary to obtain sufficient levels of intact mitochondria. An alternative method is based on digitonin-based permeabilization of cells [184]. Steroidal saponin digitonin is able to permeabilize the cytosolic membrane while leaving the mitochondrial membrane intact. After permeabilization, cytosolic metabolites can be removed by a washing step, and stable isotope tracers can be added to the isolated mitochondria to allow the subsequent analysis of the metabolome using GC-MS. A limitation of both methods is that they require extensive preparations to isolate the mitochondria, possibly interfering with their metabolic state. Nevertheless, these methods enable us to obtain information about metabolic changes at the level of the mitochondria which could potentially be lost in data representing the metabolites and metabolic pathways of the whole cell. For example, Veyrat-Durebex and colleagues used differential centrifugation to extract the fraction containing the mitochondria and endoplasmic reticulum of fibroblasts from ALS patients and controls [180]. While metabolomic analysis of whole cells showed altered levels of purine, pyrimidine, and energetic metabolisms in ALS, disturbed phosphatidylcholine levels were observed in the mitochondrial and endoplasmic reticulum fraction from ALS patients [180]. Indeed, this indicates that the metabolic defects could be compartmentalized in ALS.

The results on the mitochondrial metabolome could and perhaps should be combined with data on mitochondrial oxygen consumption to obtain feedback regarding the metabolic state of the mitochondria. Furthermore, the distribution of mitochondria across the cell is also an essential consideration in energy homeostasis. The ability of the mitochondria to traffic across the cell is based on local energy demand. The axonal transport of mitochondria, driven by on-board glycolysis, toward the energy demanding synapses is crucial for the normal function of neurons [6,185]. This process can, for instance, be measured in culture using mitochondrial tracker in a live cell imaging setup, allowing the tracking of mitochondrial transport [186,187]. An analysis of mitochondrial transport together with mitochondrial metabolism could give us a better insight on the energy supply of motor neurons in ALS.

## 4. Conclusions and Future Perspectives

In this review, we focused on tools which we believe are of particular interest in the context of ALS research. Given that disease etiology is incompletely understood for most ALS cases, omics technologies offer an advantage as these are not hypothesis driven by nature. While genomics and transcriptomics provide information about the phenotype, metabolomics is unique as it is able to link the genotype to the phenotype [188]. The compatibility of various emerging metabolomic tools with novel, single cell analysis methods and more complex patient-derived cell culture models is particularly exciting.

Indeed, great strength lies in the combination of these various existing and emerging metabolic tools to study metabolic defects. The role of energy metabolism in ALS is underappreciated and while it becomes more apparent that the mitochondrial, carbohydrate and lipid metabolism are dysregulated in ALS, the mechanisms underlying this dysregulation remain mainly unknown. Therefore, there is a need for studies applying a combination of state-of-the-art metabolic tools to unravel the possible causal link of metabolic defects to ALS pathology.

In addition, the combination of these novel metabolomic tools with new modelling strategies could increase the value of metabolic findings in ALS. Although mouse models of ALS are a valuable tool, a limitation of these current rodent models is that they are often based on the overexpression of disease-related transgenes. Indeed, overexpression of a transgene itself is proven to induce mitochondrial vacuolization and hence metabolic dysfunction [189,190]. New models, such as cultures differentiated from patient-derived iPSCs provide a great tool to investigate metabolic defects [191,192]. Isogenic controls from these patient-derived lines can be generated, hereby offering a tool to study the causative effect of mutations on the metabolism of differentiated motor neurons. Moreover iPSCs provide an excellent tool to generate 2D and 3D co-culture systems since they enable the differentiation of different cell types from a single patient [127]. Two and three dimentional ALS models containing motor neurons together with skeletal muscle fibers or endothelial cells using microdevices have recently been reported [193,194]. Adding other cells such as astocytes and oligodendrocytes could improve these model systems even further as they are known to contribute to the microenvironment of motor neurons [195,196]. Moreover, these new culture methods allow the investigation of cell–cell interactions in the context of ALS, and metabolomic analysis of co-cultures could provide valuable information about the energy metabolism of motor neurons, glia and muscle cells in a recapitulated native microenvironment. However, it will remain difficult to disentangle cell-specific signals as long as single cell metabolomics is not optimized. It should be noted that if we want to move beyond the notion of involvement of a particular cell type in motor neuron viability in the context of ALS, as established in various rodent studies, longitudinal assessment of the cellular (metabolic) phenotype throughout different stages of the disease will be required [197].

Given its unbiased nature, metabolomics could also be a valuable tool in the search for reliable biomarkers in ALS. A range of metabolic biomarkers are suggested, though no single metabolic biomarker is deemed to be reliable on its own [198]. While one metabolic biomarker might not be able to predict the disease, metabolomics offers a way to create metabolic profiles of combined biomarkers which could increase the predictive value of these markers [124]. Moreover, metabolomics could also play an important role in the search for treatment strategies. Indeed, implementation of new models and tools in ALS research offers the opportunity to bring us closer to unraveling the role of metabolism in this disease, which could enable us to design specific therapeutic strategies to interfere with the aberrant metabolism in ALS patients. To date, no efficient treatment for ALS is available and studies leading to potential new treatment strategies remain of great importance [199,200]. Efforts have been made to find new therapeutics based on the suggested metabolic defects [8,98,201,202,203,204,205], but partly due to the lack of knowledge about the metabolic defects in ALS, this search has not yet been successful. Therefore, we believe that the tools discussed in this review could play an important role in the search for new treatment strategies in ALS.

## Figures and Tables

**Table 1 genes-10-01011-t001:** Overview of technologies to analyze energy metabolism.

Technology	Analytical Platform	Measurement of	References
RNA sequencing	Numerous analytical platforms available (for review see: [206])	RNA levels of flux generating enzymes	Hrdlickova et al. [206]
Mitochondrial respiration	Seahorse XF bioanalyzer or Oxygraph-2k	Oxygen consumption rate	Nicholls et al. [74] and Hall et al. [75]
Radioactive flux tracing	^14^C and ^3^H tracing	Intracellular flux analysis	Veys et al. [81]
Metabolomics	GC-MS, LC-MS and NMR	Known metabolites	Zhou et al. [88], Fiehn et al. [89] and Edison et al. [96]
Stable isotope tracing	GC-MS, LC-MS and NMR	Enrichment of substrate to metabolome	Pinnick et al. [105], Buescher et al. [106] and Bhinderwala et al. [207] and Mrkley et al. [96]
Untargeted metabolomics	LC-MS	Known and unknown metabolites	Zhou et al. [88]
In vivo metabolomics	GC-MS, LC-MS and NMR	Metabolites in blood, urine, or tissue extracts	Broekaert et al. [137], Pinnick et al. [105], Huang et al. [150], Wang et al. [153], Roy et al. [154] and Blattman et al. [155]
Single-cell metabolomics	MALDI, LSC-MS, SIMS and LAESI	Single-cell metabolome	Laiko et al. [170], Tejedor et al. [171], Kaganman et al. [172] and Shrestha et al. [175]
Mitochondrial metabolomics	GC-MS	Mitochondrial metabolome	Gravel et al. [182] and Nonnenmacher et al. [184]

GC-MS; gas chromatography mass spectrometry, LC-MS; liquid chromatography mass spectrometry, NMR; nucleor magnetic resonance, MALDI; matrix-assisted laser desorption, LSC-MS; ive single-cell mass spectrometry, SIMS; secondary ion mass spectrometry, LAESI; laser ablation electronspray ionization.

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
