# Peer review of "Existing and Emerging Metabolomic Tools for ALS Research"

_genes, 2019, doi:10.3390/genes10121011_

Round 1

Reviewer 1 Report

Given the nature of ALS and its incompletely understood etiology, omics technologies offer an unbiased way to study the disease. In this review article, the authors focused on a subset of tools which they believe are of particular interest in the context of ALS research. Although I am not a specialist in ALS, I am familiar with all the techniques presented and found the author’s discussions of each clear and concise. Furthermore, as a non-specialist it was easy to interpret how each of these technologies would be applied in the context of ALS, and I commend the authors on their efforts. I was surprised that the authors have largely avoided discussing genomics and transcriptomics which can provide valuable phenotypic information, however, I assume this is to be outside of the scope of this review. As, the authors state metabolomics is useful to link the genotype to the phenotype, however, I found the lack of technical detail and examples somewhat disappointing. The authors finish the manuscript by discussing the compatibility of various emerging metabolomic tools with novel, single cell analysis methods and more complex patient-derived cell culture models. These prospects are particularly exciting, given the prospective nature of the review, I was happy to see their inclusion, and that they authors discussed these technologies in (relatively) greater detail. I have no issues with this manuscript being accepted in its current form, however, I feel it would greatly benefit from an additional read through and if the space allows, the inclusion of more technical discussions.

Minor Comments

My main criticism is the lack of technical detail, however, I also feel that the short paragraph from lines 75-81 does not sit well within the wider text. This may partially be a result of the first sentence, which requires rewording. However, this section would substantially benefit from further proofing, and editing for clarity.

I was happy to see the inclusion of the Seahorse assay, and also thought this was a good technology to open with. Although brief, the paragraph adequately describes the basic approach and scientific protocol, however, a two sentence explanation of the reliability and results that have been obtained with this approach would finish the paragraph without leaving the reader guessing.

Paragraphs from 211-237 appear out of place under the sub heading of stable isotope tracing

Line 342, A word appears to be missing: picked and manually guided?

Reviewer 2 Report

The manuscript submitted by Germeys et al., review the state of affairs of existing and emerging technologies and their application to ALS research. The authors carefully analyse the evolution of the metabolomic toolkit and in some cases suggest how improvements can enhance the field. Indeed, the authors describe the approaches used and enhanced to study different aspects of metabolomics and then provide a concluding summary of how the field can use such approaches. 

I have a number of recommendations which can enhance the paper.

The involvement of ALS in most of the subsections is spurious at best, using tangential links. I suggest that the authors make a more direct evaluation of how ALS is specifically linked to each subsection mentioned, given that the title is focusing on ALS. Lines 34-35 – the sentence needs to be connected in a more effective way. Introduction – My major problem with introduction is that it reads like a shopping list of potential influences of metabolism on ALS in both patients and preclinical models, and as such lacks clear direction and hypothesis. I suggest the authors give their opinion more clearly throughout the introduction. Introduction – Given the hot topic that is the microbiome - what are the authors thoughts on its impact on ALS? Introduction – Is there any preclinical evidence demonstrating dysregulated energy metabolism? Introduction - What are the authors thoughts on selective vulnerability of MNs? ie. extraocular vs limb muscles? gamma vs alpha? Introduction - What about the role of lipid metabolism in ALS and vulnerability? see 10.3389/fnmol.2018.00010 Introduction – The authors are missing discussing an important paper. 1080/21678421.2019.1621346 Section 2 - Please provide an introductory statement(s) in section 2 and before 2.1 Section 2.2 - How does the following study fit in ? 10.1113/JP272988 Section 2.2, lines 138-141 - What are the outcomes of those studies linked to ALS mutations? This is pertinent information given the title and ALS-focus of the review. Please expand. Section 2.4 - The metabolomics paragraph is imbalanced compared to the rest of section 2. A lot is superfluous and can be found in textbooks and hence should be reduced. Section 2.4 - What about biomarkers in metabolomics? How does this fit in with your hypothesis? See 10.3389/fneur.2019.00191 Section 2.5 - The authors are not including pertinant data from the following paper - 10.1136/jnnp-2017-317887 Section 3.1 – Lines 290-293 – The following review is most up to date on animal models of ALS. - 10.1242/dmm.037424 Section 3.2 – Lines 322-325 – Such as...? Please extrapolate. This section needs to be significantly expanded. Section 3.3 - How would one differentiate changes in single cell metabolomics as a result of ALS-toxicity from standard in vitro cell death? Section 3.4 – Lines 466-468 - The following reference might be useful here 10.1073/pnas.1605210113

Round 2

Reviewer 2 Report

Thank you for addressing my concerns.